# *N*-Phenacyldibromobenzimidazoles—Synthesis Optimization and Evaluation of Their Cytotoxic Activity

**DOI:** 10.3390/molecules27144349

**Published:** 2022-07-07

**Authors:** Anna Kowalkowska, Konrad Chojnacki, Maciej Multan, Jan K. Maurin, Edyta Łukowska-Chojnacka, Patrycja Wińska

**Affiliations:** 1Faculty of Chemistry, Warsaw University of Technology, Noakowskiego St. 3, 00-664 Warsaw, Poland; konrad.chojnacki@pw.edu.pl (K.C.); maciek.multan@gmail.com (M.M.); edyta.chojnacka@pw.edu.pl (E.Ł.-C.); patrycja.winska@pw.edu.pl (P.W.); 2National Medicines Institute, Chełmska St. 30/34, 00-725 Warsaw, Poland; j.maurin@nil.gov.pl; 3National Centre for Nuclear Research, Andrzeja Sołtana St. 7, 05-400 Otwock, Poland

**Keywords:** dibromobenzimidazoles, *N*-phenacyl, synthesis optimization, *N*-alkylation, cytotoxicity, cancer, apoptosis, acute lymphoblastic leukemia

## Abstract

Antifungal *N*-phenacyl derivatives of 4,6- and 5,6-dibromobenzimidazoles are interesting substrates in the synthesis of new antimycotics. Unfortunately, their application is limited by the low synthesis yields and time-consuming separation procedure. In this paper, we present the optimization of the synthesis conditions and purification methods of *N*-phenacyldibromobenzimidazoles. The reactions were carried out in various base solvent-systems including K_2_CO_3_, NaH, KOH, *t-*BuOK, MeONa, NaHCO_3_, Et_3_N, Cs_2_CO_3_, DBU, DIPEA, or DABCO as a base, and MeCN, DMF, THF, DMSO, or dioxane as a solvent. The progress of the reaction was monitored using HPLC analysis. The best results were reached when the reactions were carried out in an NaHCO_3_–MeCN system at reflux for 24 h. Additionally, the cytotoxic activity of the synthesized compounds against MCF-7 (breast adenocarcinoma), A-549 (lung adenocarcinoma), CCRF-CEM (acute lymphoblastic leukemia), and MRC-5 (normal lung fibroblasts) was evaluated. We observed that the studied cell lines differed in sensitivity to the tested compounds with MCF-7 cells being the most sensitive, while A-549 cells were the least sensitive. Moreover, the cytotoxicity of the tested derivatives towards CCRF-CEM cells increased with the number of chlorine or fluorine substituents. Furthermore, some of the active compounds, i.e., 2-(5,6-dibromo-1*H*-benzimidazol-1-yl)-1-(3,4-dichlorophenyl)ethanone (**4f**), 2-(4,6-dibromo-1*H*-benzimidazol-1-yl)-1-(2,4,6-trichlorophenyl)ethanone (**5g**), and 2-(4,6-dibromo-1*H*-benzimidazol-1-yl)-1-(2,4,6-trifluorophenyl)ethanone (**5j**) demonstrated pro-apoptotic properties against leukemic cells with derivative **5g** being the most effective.

## 1. Introduction

The *N*-phenacylazole substituent is found in many biologically active compounds exhibiting antifungal [1,2,3,4,5,6], antioxidant [5,7], antibacterial [8,9,10], and anticancer activity [9,11]. This substituent can also be easily transformed to an analog with a hydroxyl group, which is a known privileged scaffold in many antifungal drugs, e.g., fluconazole, voriconazole, isavuconazonium sulfate, efinaconazole, and isavuconazole [12,13]. Other similar compounds, such as albaconazole, are under investigation [12]. The search for new antimycotics is necessary due to the increasing resistance of fungal pathogens to the available drugs, the toxicity of these drugs, and their undesirable interactions. Moreover, problems have arisen that involve increased mortality from fungal infections among immunocompromised patients and higher healthcare costs [14,15,16,17,18,19,20,21,22,23,24,25,26,27,28].

Recently we described the synthesis and antifungal activity against *Candida albicans* and *Candida neoformans* of *N*-phenacyl-4,6/5,6-dibromobenzimidazoles [29]. These compounds were obtained by *N*-alkylation of appropriate dibromobenzimidazoles with 10 different phenacyl halides. All reactions were carried out in a K_2_CO_3_–MeCN system. Depending on the alkylating agent, the reaction yields ranged from 13% to 94%. The highest values were obtained in reactions of 4,5/4,6-dibrombenzimidazole with phenacyl bromide, 4-fluoro-, 4-chloro-, and 4-bromophenacyl chloride, whereas the lowest were obtained with the respective chloroketones substituted with two or three chlorine or fluorine atoms (13–24%). The biological study and analysis of the structure–activity relationship proved that the synthesized compounds exhibited significant antifungal activity, especially 4,6-dibromobenzimidazole containing a difluoro- or dichloro-substituted benzene ring in the phenacyl moiety. Unfortunately, these compounds were obtained in the lowest yields (15–22%). In order to use them as the lead structures in the synthesis of new antifungal agents, the development of an efficient method for their synthesis is necessary. The literature data indicate efficient methods for the alkylation of benzimidazole with monohalogenophenacyl halides, but there is little information on the corresponding reactions with di- or trihalogenophenacyl chlorides. For instance, the reaction of benzimidazole with phenacyl bromide is efficiently carried out (yield up to 92%) using NEt_3_ as a base in acetone [30] or MeCN [31,32] at reflux, in some cases with the addition of a catalytic amount of tetrabutylammonium bromide (TBAB) [33,34] or tetrabutylammonium hydrogen sulfate (TBAHSO_4_) [35]. The reactions carried out without the addition of an external base reached only 53–60% yield [36,37]. The same reaction was performed in a KOH–EtOH system under microwave irradiation [38] or a K_2_CO_3_–DMF system under nitrogen at rt [7] (the yield not given). Moreover, *N*-alkylation of benzimidazole with 4-chlorophenacyl chloride was effectively carried out in a K_2_CO_3_–CH_2_Cl_2_ system (yield 75%) [39]. The US-assisted *N*-alkylations of benzimidazole were carried out using 0.1 equivalent of an NEt_3_–acetone system, providing six *N*-phenacylbenzimidazoles in 90–95% yields [1,40]. The alkylation of 4- and 2,4-disubstituted benzimidazoles was carried out using 2-chloroacetophenone in an NaH–DMF system under argon atmosphere at 30 °C giving mixtures of isomeric products [41]. 5-Cyanobenzimidazole was effectively alkylated with 2,4′-dibromoacetophenone in an NaH–THF system at reflux resulting in the mixture of regioisomers obtained in a 90% yield [42]. The alkylation of 2-methyl-, 2-ethyl-, 2-isopropyl-, and 5,6-dimethylbenzimidazole with 2,3′-dibromoacetophenone in the K_2_CO_3_–DMF system at 80 °C afforded products in moderate yields (32–55%) [43].

The alkylation of benzimidazole with 2,4-dichlorophenacyl chloride was effectively performed in solvent-free conditions by heating appropriate α-chloroketone with an excess of benzimidazole using conventional heating (yield 88%) or MW (yield 93%) [44], whereas the same reaction in a K_2_CO_3_–cat. KI–MeCN system at reflux afforded products with a 58% yield [45]. To the best of our knowledge, these are the only examples of benzimidazole *N*-alkylation with phenacyl halides described in the literature, except for the reactions of benzimidazole and its 2- or 5,6-substituted derivatives with 4- or 5-(bromoacetyl)-2-chlorobenzenesulfonamide and 5-(bromoacetyl)-2,4-dichlorobenzenesulfonamide carried out in an NaOAc–THF system at rt [46,47,48]. However, there are well-known reactions of 2,4-dichloro- and 2,4-difluorophenacyl chlorides with 1,2,4-triazole and imidazole. The appropriate *N*-phenacyltriazoles/imidazoles were usually obtained in moderate to good yields (39–98%) using the K_2_CO_3_–MeCN system at reflux [49,50,51,52,53] with the addition of the catalytic amount of KI [45]. When these reactions were carried out in the same system (K_2_CO_3_–MeCN) without KI at room temperature, divergent yields were reported (27–85%) [54,55]. Moreover, both mentioned azoles can be *N*-alkylated at reflux in an NaHCO_3_-toluene system (yields 42–87%) [4,56,57,58,59,60,61,62], NEt_3_-DMF [63], or NEt_3_-MeOH [64]. Triazole can also be efficiently alkylated with 2,4-difluorophenacyl chloride in an K_2_CO_3_–CH_2_Cl_2_ system at rt (yield of 70%) [65,66,67].

Considering the antifungal potential of *N*-phenacyldibromobenzimidazoles, we decided to continue the research on their synthesis and to develop the reaction conditions that would allow the simplification of the procedure of their isolation and obtain the title compounds in satisfactory yields. In addition, for the first time we evaluated the cytotoxic effect of the synthesized compounds [68] on three tumor cell lines, i.e., MCF-7 (breast adenocarcinoma), A-549 (lung adenocarcinoma), CCRF-CEM (acute lymphoblastic leukemia), and MRC-5 (normal lung fibroblasts).

## 2. Results and Discussion

### 2.1. Synthesis of Title Compounds

*N*-Phenacyldibromobenzimidazoles **4**–**5a**–**j** were obtained by *N*-alkylation of 5,6- or 4,6-dibromobenzimidazole (**1** or **2**) with various phenacyl halides **3a**–**j**. The reaction conditions were optimized using a model reaction, *N*-alkylation of 5,6-dibromobenzimidazole **1** with 2,4-dichlorophenacyl chloride **3e** (Figure 1). The optimized parameters were: the type of solvent and base, the molar ratio of substrates, the temperature, and the time of the reaction. The conversion of each reaction was controlled by HPLC analysis. We started our investigations by evaluating the reaction of **1** with **3e** in the K_2_CO_3_–MeCN system, using various molar ratios of the substrates (Table 1, Entry 1–6). The best result was observed for a 3-fold excess of benzimidazole **1** over the alkylating agent **3e** (Table 1, Entry 1). The yield of the reaction reached only 28%. Subsequent experiments were carried out using other base–solvent systems at rt for 24 h (Table 1, Entry 7–23). The product **4e** was detected using the K_2_CO_3_–dioxane (yield of 7%, Table 1, Entry 16), Et_3_N–MeCN, Et_3_N–DMF or DIPEA–MeCN systems (yield of 15–22%, Table 1, Entry 20–22). When the reaction was conducted in the presence of NaHCO_3_ in MeCN (Table 1, Entry 18) the yield was very low (<1%) and alkylating agent **3e** remained unreacted. In other systems, despite low product **4e** yields, high chloroketone conversion was observed. To increase the yield of the reaction, the following experiments (Table 1, Entry 24–27) were carried out at reflux for 3 h. Depending on the conditions, the product was obtained in yields ranging from 28% to 44%. The highest values were reached using the NaHCO_3_–MeCN system. Considering the results of further experiments (Table 1, Entry 28–31), the NaHCO_3_–MeCN system was chosen for optimization of the molar ratio of substrate **1**: alkylating agent **3e**: base. The highest yield (54%) was observed using the NaHCO_3_–MeCN system, with 0.3 mol excess of α-chloroketone **3e** at reflux for 24 h (Table 1, Entry 34).

Further optimization of the reaction conditions was carried out on a preparative scale (Table 2) and yields of the reactions were calculated after the purification of the product.

The reaction of equimolar amounts of compounds **1** and **3e** with 20 equiv. of NaHCO_3_ in 50 mL of MeCN at reflux afforded product **4e** with the yield of 58% (after column chromatography), while at 60 °C the yield was reduced by 20% (Table 2, Entry 1–2). When the molar ratio of compound **3e**:**1** increased to 1.3:1 or the volume of the solvent was reduced to 30 mL, the yield of **4e** was not improved (Table 2, Entry 3,4). Further optimization was performed with a lower excess of NaHCO_3_ (5 equiv.). In all cases (Table 2, Entry 5–9) the reactions were initially carried out with equimolar amounts of substrates. The additional portions of chloroketone **3e** were added gradually, after 17 h and 20 h of reaction time. This method allowed us to receive product **4e** in a yield of 67% (after column chromatography, Table 2, Entry 5). Further investigation of the product separation method revealed that good results were obtained by treating the crude product with a small volume of ethyl acetate and filtering the precipitated solid. This procedure allowed us to obtain product **4e** in a 63–66% yield, using 50 and 40 mL of MeCN, respectively (Table 2, Entry 6–7). A further decrease in solvent volume to 30 and 20 mL did not improve the results. In the first case, product **4e** contained some impurities (Table 2, Entry 8), while in the case of the lowest volume of solvent only traces of the product were precipitated when ethyl acetate was added (Table 2, Entry 9).

Additionally, one more base–solvent system was investigated. Carrying out the reaction of equimolar amounts of substrates **1** and **3e** in the presence of K_2_CO_3_ in MeCN at reflux for 0.5 h, product **4e** was afforded in a yield of 61%. This value was obtained when the product was purified by column chromatography. In this case, the precipitation method failed (Table 2, Entry 10–11).

With the optimal conditions in hand, we carried out the reactions of dibromobenzimidazoles **1** and **2** with phenacyl chlorides **3e**–**j** in the NaHCO_3_–MeCN system, at reflux, with benzimidazole/alkylating agent ratio 1/1 at the beginning. Further portions of chlorides **3** were added during the reaction. In most cases full conversion was observed after 24 h, using 1.5–2 equiv. of chloroketone **3**. Usually, *N*-alkylation of 4,6-dibromobenzimidazole required smaller excesses of alkylating agent **3**. Most of the crude products were precipitated from oils obtained after evaporation of the filtered reaction mixtures. In the case of 2,4,6-trifuoroderivatives **4j**,**5j** and 2,4-difluoroderivative **4h**, products were isolated by column chromatography.

The only exceptions were the *N*-alkylation of benzimidazoles **1**,**2** with 2,4,6-trichlorophenacyl chloride **2g**. In the NaHCO_3_–MeCN system, no product was formed. The reactions were carried out in the presence of K_2_CO_3_ in MeCN at reflux. Products **4g** and **5g** were isolated by column chromatography due to the formation of large amounts of by-products.

To compare the effect of base–solvent systems, the respective reactions of benzimidazoles **1**,**2** with phenacyl bromide **3a** and their monosubstituted derivatives **3b**–**d** were performed. In the case of 4,6-dibromobenzimidazole **2**, a nearly full conversion after 16–20 h, using 1.05–1.25 equivalent of alkylating agent **3a**–**d** was observed. The yields of compounds **5a**–**d** exceeded 80%. Meanwhile, the rate of 5,6-dibromobenzimidazole **1** alkylation with **3a**–**d** was significantly slower (after carrying out the reaction for 24 h with two equivalents of **3a**–**d** added in portions, the full conversion of substrate **1** was often not observed). As a result, the yields of compounds **4a**–**d** were lower than those obtained in the K_2_CO_3_–MeCN system at rt [29]. A double decrease in acetonitrile volume mainly resulted in the formation of more by-products, but it did not allow the reduction in the reaction time (Figure 2 and Figure 3, Table 3).

In the case of the alkylation of 4,6-dibromobenzimidazole, the formation of two isomeric products, having bromine atoms at 4,6 or 5,7 positions, is possible. The formation of the 4,6-isomer was unambiguously confirmed by the X-ray crystallography of compound **5d** (Figure 1).

To explain the low synthesis yields of compounds **4** and **5** in the K_2_CO_3_–MeCN system, we carried out the reaction of the model substrate, 2,4-dichlorophenacyl chloride **3e**, under these conditions, for 24 h at rt. We observed nearly a full conversion of chloroketone **3e** and the formation of a complex mixture of products with similar *R_f_* values. Three of these products were isolated. Based on NMR spectra, HRMS analysis, and the literature data [69,70,71,72,73,74], one of the compounds was assigned the structure of chloromethyl oxirane **6e**. For the other two separated products, the structures were not unambiguously assigned. For one of these products, the structure of diepoxide **7e** can be proposed (Figure 4). Similar structures were reported earlier [73,75,76].

In the case of the last product, containing nine aromatic and three aliphatic protons, the formation of *trans*-1,2,3-tribenzoylcyclopropane **8e** was suggested. The analysis of aromatic proton signals indicated the presence of two identical benzene rings. On the other hand, for such compounds, the presence of a doublet and triplet in the proton spectrum in the aliphatic range is characteristic. [71,77,78]. However, a doublet and a doublet of doublets are present in the spectrum of the isolated compound. Previous literature data indicate that analogous compounds, with unsubstituted benzene rings, can be formed in reactions of phenacyl chloride under alkaline conditions [73,74,79,80].

### 2.2. Evaluation of Biological Activity

To evaluate the cytotoxic activity of compounds **4a**–**j** and **5a**–**5j**, we performed an MTT test for three tumor cell lines, i.e.,: MCF-7 (breast adenocarcinoma), A-549 (lung adenocarcinoma), CCRF-CEM (acute lymphoblastic leukemia), and one normal cell line, MRC-5 (normal lung fibroblasts). The EC_50_ values, describing the half maximal effective concentration of each tested compound, were calculated and are summarized in Table 4. The representative sigmoidal dose–response curves are shown in Appendix A (see Appendix A).

The study showed that most of the tested derivatives exhibited biological activity, with the exception of derivative **4d** (4-Br), which was sparingly soluble in DMSO. Interestingly, derivative **5d** (4-Br) demonstrated moderate activity against all the tested cell lines, with the lowest value of EC_50_, 27.75 µM for MCF-7. The sensitivity of the studied cell lines was different, and the MCF-7 line shows the greatest sensitivity to the tested compounds, with the lowest EC_50_ for derivative **5g** (2,4,6-Cl_3_) (23.98 μM). This compound also showed the best activity against the CCRF-CEM line with an EC_50_ of 26.64 μM; however, it was also cytotoxic to normal cells (MRC-5) with an EC_50_ of 26.9 μM. Among the studied cell lines, A-549 cell line appeared to be the least sensitive to the tested compounds, with compound **4i** (2,5-F_2_) showing the most activity with an EC_50_ of 37.87 μM. Interestingly, we observed that as the number of chlorine or fluorine substituents in 4,6-dibromobenzimidazole derivatives increased, their cytotoxicity was better towards CCRF-CEM cells, i.e., from the least cytotoxic **5b** (4-F) and **5c** (4-Cl) to the most active **5j** (2,4,6-F_3_) and **5g** (2,4,6-Cl_3_), respectively. However, among the tested compounds, the highest selectivity index was obtained for **5a** (2.04) for MCF-7, respectively.

In order to evaluate the pro-apoptotic properties of the most active derivatives towards CCRF-CEM, i.e., **4f** (3,4-Cl_2_), **5g** (2,4,6-Cl_3_), and **5j** (2,4,6-F_3_), we determined annexin V-binding to phosphatidylserine in CCRF-CEM cells by means of flow cytometry. The obtained results indicated that all tested compounds used at 45 µM concentration induced apoptosis effectively in CCRF-CEM (Figure 2) with the highest percent of apoptotic cells, i.e., 75.6% obtained after treatment with **5g** (Figure 2b). This compound also induced apoptosis at a 30 µM concentration, giving 32% of cells in late apoptosis (Figure 2b).

## 3. Materials and Methods

Commercially available reagents from Sigma Aldrich (Darmstadt, Germany) and Avantor (Gliwice, Poland) were used as supplied. Solvents: DMF, THF, DMSO, and MeCN (for reaction with NaH) were dried with standard methods. Thin-layer chromatography was carried out on TLC aluminum plates with silica gel Kieselgel 60 F_254_ (Merck, Darmstadt, Germany) (0.2 mm thickness film). The column chromatography was performed using Silica gel 60 (Merck, Darmstadt, Germany) of 40–63 μm.

Dimethyl sulphoxide (DMSO), Molecular Biology grade, used as a solvent for all stocks of the chemical agents, was obtained from Roth (Karlsruhe, Germany). All reagents used in flow cytometry analysis were purchased from BD Biosciences Pharmingen (San Diego, CA, USA).

HPLC analyses were performed on PE NELSON NCI900 chromatograph (Waltham, MA, USA) equipped with a UV-VIS (Perkin Elmer 785A detector, Waltham, MA, USA) and a reverse-phase column Hichrom HI-5C18-3959 (250 × 1.0 mm, 5 µm), 10 µL of the sample was injected. A gradient method comprising water/acetonitrile was applied as follows: 0–20 min 20/80 (1 mL/min), 20–25 min 20/80–0/100 (1 mL/min), 25–60 min 0/100 (1 mL/min). UV–VIS detector was set at a wavelength of 254 nm.

The melting points were measured with a commercial apparatus Thomas-Hoover “UNI-MELT” on samples placed in glass capillary tubes and were not corrected. The ^1^H and ^13^C NMR spectra were measured with a Varian 500 spectrometer operating at 500 MHz for ^1^H and 125 MHz for ^13^C nuclei. Chemical shifts (δ) are given in parts per million (ppm) relative to the residual solvent signal (CDCl_3_, δ_H_ of residual CHCl_3_ 7.26 ppm); signal multiplicity assignment: s, singlet; br s, broad singlet; d, doublet; dd, doublet of doublets; m, multiplet; coupling constants (J) are given in hertz (Hz). All these measurements were made at Warsaw University of Technology. High-resolution mass spectrometry (HRMS) was carried out on Q Exactive Hybrid Quadrupole–Orbitrap Mass Spectrometer (Bremen, Germany), ESI (electrospray) with spray voltage 4.00 kV at Institute of Biochemistry and Biophysics Polish Academy of Science (IBB PAS, Warsaw, Poland. The most intensive signals are reported.

Analytical data of compounds **4a**, **4j**, **5b**, **5d**, **5e**, **5f**, **5h**, **5j** [29], and **4b** [82] were described earlier.

### 3.1. Procedure for Optimization Reactions in NaHCO_3_-MeCN System

To the 5,6-dibromobenzimidazole **1** (1 mmol, 276 mg) in MeCN (20–50 mL, see Table 2) 2,4-dichlorophenacyl chloride **3e** (amount: see Table 2, Entry 1–9) was added, followed by NaHCO_3_ (5 or 20 mmol, see Table 2). The reaction was heated to 60 °C or reflux and stirred for 24 h, cooled, filtered through a short pad of celite, washed with MeCN (4 × 5 mL), evaporated to dryness. The residue was purified by column chromatography (silica gel, eluent: CHCl_3_, Entry 1–6) or treated with EtOAc to precipitate product **4e** (Entry 7–10). The precipitated compound **4e** was filtered and washed with cold EtOAc.

### 3.2. Procedure for Optimization Reactions in K_2_CO_3_-MeCN System

To the refluxed solution of 5,6-dibromobenzimidazole **1** (1 mmol, 276 mg) and 2,4-dichlorophenacyl chloride **3e** (1 mmol, 223 mg) in MeCN (25 mL) K_2_CO_3_ (5 mmol) was added. The reaction refluxed for 0.5 h, cooled, filtered through a short pad of celite, washed with MeCN (4 × 5 mL), evaporated to dryness. The residue was purified by column chromatography (silica gel, eluent: CHCl_3_, Entry 11, 61% yield of **4e**) or treated with EtOAc (traces of precipitated **4e**) (see Table 2, Entry 10–11).

### 3.3. Synthesis of N-Phenacylbenzimidazoles ***4***,***5a***–***f***, ***4***,***5h***–***j***

The mixture of benzimidazole **1** or **2** (4 mmol), MeCN (160 mL), phenacyl halide **3** (4 mmol), and NaHCO_3_ (20 mmol, 1.68 g) was stirred magnetically at reflux. The reaction was monitored by TLC (CHCl_3_/MeOH 95/5 *v*/*v*) typically after 16, 20, and 24 h. If necessary, additional portions of phenacyl halide (1–4 mmol) were added after 16.5 and 20.5 h. In most cases, full conversion of benzimidazole **1** or **2** was observed after ca 24 h. The mixture was cooled, filtered through a short pad of celite, washed with MeCN (4 × 10 mL), evaporated to dryness. The solid residue was transferred to a Schott funnel and washed with the respective solvent (5–7 × 2–10 mL, see Appendix A). The oily residue was purified by column chromatography (silica gel, CHCl_3_ as eluent). If full conversion of benzimidazole **1** or **2** was not observed after 24 h, further portions of substrate **3** were added, the reaction was continued and worked-up as described above.

Traces of isomeric 2-(5,7-dibromo-1*H*-benzimidazol-1yl)-1-arylethanones formed in *N*-alkylations of 4,6-dibromobenzimidazole **2** with phenacyl halides **3a**–**j** were removed by column chromatography (silica gel, chloroform as eluent) or by crystallization (MeOH or EtOH).

### 3.4. Synthesis of N-(2,4,6-Trichlorophenacyl)Benzimidazoles ***4g***,***5g***

The mixture of benzimidazole **1** or **2** (4 mmol), MeCN (160 mL), phenacyl halide **3g** (4 mmol), and K_2_CO_3_ (20 mmol, 2.76 g) was stirred magnetically at reflux. The reaction was monitored by TLC (CHCl_3_/MeOH 95/5 *v*/*v*). Additional portions of phenacyl halide (1 mmol each one) were added after 2, 3, 4, 5, 6, 7 h. After 8 h the mixture was cooled, filtered through a short pad of celite, washed with MeCN (4 × 10 mL), evaporated to dryness. The residue was purified by column chromatography (silica gel/chloroform, eluent: chloroform). The fractions containing the product were collected, evaporated, treated with Et_2_O (5 mL), filtered, and washed with Et_2_O (5 × 3–4 mL).

### 3.5. Cell Culture and Agents Treatment

CCRF-CEM (ECACC 85112105) human Caucasian acute lymphoblastic leukaemia and MRC-5 pd30 Human fetal lung fibroblasts (ECACC 05090501) were purchased from European Collection of Authenticated Cell Cultures, whereas A-549 (ATCC CCL 185) human lung carcinoma and MCF-7 (ATCC HTB-22) human Caucasian breast adenocarcinoma cell line were purchased from American Type Culture Collection. A-549 and MCF-7 cell lines were cultured in high glucose DMEM (Biowest) supplemented with 10% fetal bovine serum (Biowest), 2 mM L-glutamine and antibiotics (100 U/mL penicillin, 100 µg/mL streptomycin). CCRF-CEM were cultured in RPMI 1640 supplemented with 10% fetal bovine serum (EuroClone), 2 mM L-glutamine and antibiotics (100 U/mL penicillin, 100 µg/mL streptomycin). MRC-5 pd30 human fibroblasts (ECACC) were cultured in MEME, Minimum Essential Medium Eagle (Merck) supplemented with 10% fetal bovine serum (Merck), 2 mM l-glutamine, antibiotics (100 U mL^−1^ penicillin, 100 μg mL^−1^ streptomycin, Merck) and 1% non-essential amino-acids (Merck). Cells were grown in 75 cm^2^ cell culture flasks (Sarstedt, Nümbrecht, Germany), in a humidified atmosphere of CO_2_/air (5/95%) at 37 °C. All the experiments were performed in exponentially growing cultures. Stock solution of tested compounds were prepared in DMSO and stored in −20 °C for maximum one month. For the cytotoxicity studies, stock solutions were diluted 200-fold with the proper culture medium to obtain the final concentrations. For cytotoxicity studies 2-fold serial dilutions of the tested compounds were prepared in the proper medium in the range from 3.125 µM to 200 µM.

### 3.6. 3-(4,5-Dimethylthiazol-2-yl)-2,5-Diphenyltetrazolium Bromide (MTT)-Based Viability Assay

After incubation with the test compounds, MTT test was performed as described previously [83]. Optical densities were measured at 570 nm using BioTek microplate reader. All measurements were carried out in a minimum of three replicates.

### 3.7. Detection of Apoptosis by Flow Cytometry

CCRF-CEM cells were seeded in 24-well plates at 2 × 10^5^ cells/well and treated with the tested compounds used in 15 µM, 30 µM, and 45 µM concentrations. After exposure to the examined compounds, the cells were collected, centrifugated at 200× *g* at 4 °C for 5 min, washed twice with cold phosphate–buffered saline (PBS), and subsequently suspended in binding buffer. Subsequently, 100-μL aliquots of the cell suspension were labelled according to the kit manufacturer’s instructions. Briefly, annexin V-fluorescein isothiocyanate, and propidium iodide (BD Biosciences, Pharmingen, San Diego, CA, USA) were added to the cell suspension, and the mixture was vortexed and then incubated for 15 min at RT in the dark. A cold binding buffer (400 μL) was then added and the cells were vortexed again and kept on ice. Flow cytometric measurements were performed within 1 h after labeling. Viable, necrotic, early, and late apoptotic cells were detected by FACSCanto II flow cytometer (BD Biosciences, San Diego, CA, USA) and analysed using BD FACSDiva software.

## 4. Conclusions

In summary, the above results indicate that *N*-phenacyldibromobenzimidazoles can be efficiently synthesized in different base–solvent systems depending on the substitution pattern in the phenacyl moiety. All compounds, substituted with two chlorine **4**,**5e**,**f** or fluorine atoms **4,5h,i**, as well as trifluoroderivatives **4**,**5j**, are easily obtained in the NaHCO_3_–MeCN system at reflux, while synthesis of the most sterically crowded trichloroderivatives **4**,**5g** required K_2_CO_3_ instead of NaHCO_3_. On the other hand, unsubstituted **5a** or monosubstituted **5b**–**d** derivatives of 4,6-dibromobenzimidazole are efficiently synthesized in both the K_2_CO_3_–MeCN system at rt and the NaHCO_3_–MeCN system at reflux, while for isomeric 5,6-dibromoderivatives **4a**–**d**, the first system is the best choice.

Taking into account the results of cytotoxicity studies, we conclude that the introduction of chlorine or fluorine substituents into 4,6-dibromobenzimidazole derivatives increases their cytotoxicity towards leukemic cells. On the one hand, the derivatives demonstrate some pro-apoptotic properties, which is an important feature of potential anticancer drugs; however, they are also cytotoxic to normal cells.

Further modifications and synthetic applications of the title compounds as well as evaluation of their biological activity are under investigation. 

## Data Availability

Not applicable.

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
