# Peer review of "N-Phenacyldibromobenzimidazoles—Synthesis Optimization and Evaluation of Their Cytotoxic Activity"

_molecules, 2022, doi:10.3390/molecules27144349_

Round 1

Reviewer 1 Report

Title: N-Phenacyldibromobenzimidazoles – synthesis optimization and evaluation of their cytotoxic activity

In this research article, the authors have reported the optimization of synthesis conditions and purification methods of N-phenacyldibromobenzimidazoles. The authors have carried out this reaction in various base-solvent system and found that NaHCO3-MeCN system at reflux for 24 h gave best result. The progress of this reaction was monitored by HPLC analysis. Also, the authors have evaluated the cytotoxic activity of the synthesized compounds against MCF-7 (breast adenocarcinoma), A-549 (lung adenocarcinoma), CCRF-CEM (acute lymphoblastic leukemia) and MRC-5 (normal lung fibroblasts). Based on the cytotoxicity studies, the authors have concluded that the introduction of chlorine or fluorine substituents into 4,6-dibromobenzimidazole derivatives increases their cytotoxicity towards leukemic cells. On the one hand, the derivatives demonstrate some pro-apoptotic properties, which is an important feature of potential anticancer drugs. The simplicity of this chemistry would make this approach a practical one for adaptation by other research groups. Based on the importance of this work, I suggest publishing in “Molecules” after minor revision. Several suggestions are made for the minor revision.

1)     In the supporting information, Chemical Shift d, coupling constant J, Rf should be in italic.

2)     Replace DMSO-d6 with DMSO-d6 in the spectral data for all compounds.

3)     For all the synthesized compound, provide the Br isotope [M + 2]+ mass in HRMS data

4)     If possible, try to give the full structures for the synthesized compounds in the manuscript.

Author Response

Ad.1 . In the supporting information, indicated symbols (δ, J, Rf) have been written in italics.

Ad. 2. DMSO-d6 was replaced with DMSO-d6 in the spectral data for all compounds.

Ad. 3. The Br isotope masses [M +2]+ as well as [M-2]+ were added for all compounds 4 and 5.

Ad. 4. The full structures for the synthesized compounds were added in Scheme 3 (below Table 3 in the manuscript).

Reviewer 2 Report

The optimization of the reactions conditions was very obvious. No different heating method (microwave, por example) was used, no organic bases were used, no anhydrous solvents were used, etc. In general, the described optimization is a organic synthesis routine process which does not require a publication of these results.

Cytotoxic activity studies show acceptable but not very good results.

Author Response

We agree with the reviewer that the optimization is obvious. However, we used some organic bases (DBU, Et3N, DIPEA, DABCO – Table 1, Entry 19-23, 26-28, 30-31) as well as anhydrous solvents (DMF, THF, DMSO, and MeCN (for reaction with NaH)). We apologize for the lack of this information in the first version of the manuscript, so the sentence: “Solvents: DMF, THF, DMSO, and MeCN (for reaction with NaH) were dried with standard methods” has been added to the description in Material and Methods.

We realize that the cytotoxic activity studies do not show excellent bioactivity results, so our further studies will focus on chemical modifications of the title compounds.

Reviewer 3 Report

The manuscript submitted by Kowalkowska et al presented systematic optimization of synthesis conditions including base, temperature, time, and the ratio between reagents for the N-Phenacyldibromobenzimidazoles derivatives which were initially studied by the same group before. Cytotoxicity of all compounds against three tumor cell lines and one normal cell line was tested.  There are the following issues that need to be addressed:

1. The cytotoxicity results (Ec50 value) were over-interpreted. The largest difference between the normal cell line MRC-5 vs a tumor cell line (MCF-7) is 57.92 vs 28.35 (SI = 2.04). I don't think there could be any users based on this kind of difference. Also there are no errors for all Ec50 values.

2. For biological activity evaluation, a positive control is always required. In both cytotoxicity testing against four cell lines and the determining annexin V-binding to phosphatidylserine in CCRF-CEM cells, there is no positive control.

3. A synthesis scheme of the model reaction between 1 and 3e should be added before table 1.

4. Table 1, entry 34 & 35, the ratio should be '1 : 1.3 : 20' and '1 : 1.6 : 20' instead of  '1 : 1,3 : 20' and '1 : 1,6 : 20'.

5. Table 3, entry 4, 'c' should be 'd'.

6. Chemical structures should be added to all NMR spectra in supporting information.   

Author Response

Ad.1. We agree with this comment, that is why the conclusions include the following sentences "On the one hand, the derivatives demonstrate some pro-apoptotic properties, which is an important feature of potential anticancer drugs, on the other hand, they are also cytotoxic to normal cells. Further modifications and synthetic applications of the title compounds as well as evaluation of their biological activity are under investigation."  SD values were added to all EC50.

Ad. 2. We added EC50 values for commercially available inhibitor of protein kinase CK2, i.e. CX-4945 (Silmitasertib), Table 4.

Ad. 3. A synthesis scheme of model reaction between compounds 1 and 3e has been added before Table 1.

Ad. 4. In Table 1, entry 34 and 35 commas have been replaced by periods.

Ad. 5. In Table 3, entry 4, "c" has been replaced by "d".

Ad. 6. Chemical structures of compounds 4 and 5 have been added to NMR spectra (both description  and copies of the spectra).

Reviewer 4 Report

The manuscript describes the optimisation of synthesis and purification of N-phenacyl derivatives of 4,6- and 5,6-dibromobenzimidazoles and their cytotoxic activity on different cancer cell lines, MCF-7, A-549, 21 CCRF-CEM, and normal fibroblasts (MRC-5). The main aim of this study was to increase the yields of the reaction by which these compounds are obtained and to facilitate their isolation and purification. Since the results of their antifungal activity have already been published in a previous paper, their cytotoxicity on cancer cells and normal fibroblasts was examined here for the first time. The influence of the ratio of reactants and base, the choice of solvent and base, reaction time and temperature was systematically monitored and recorded (HPLC chromatograms of optimization reactions are included as a supplement info). Different methods for product separation were also investigated. The obtained compounds were characterized by 1H and 13C NMR, and HRMS. Furthermore, X-ray analysis was performed for compound 5d to confirm the structure. Products of 2,4-dichlorophenacyl chloride 3e self-condensation reaction were isolated and characterised, and mechanism for their formation is proposed in the manuscript.

In the assessment of cytotoxic activity of the compounds on the aforementioned cell lines, MTT test was performed, and annexin V-FITC staining was used to evaluate pro-apoptotic properties by flow cytometry.

The paper is well organized, the results are presented in details and well discussed, the materials and methods are sufficiently described. In conclusion, the most successful approaches in the synthesis and purification of the desired products and cytotoxic activity are highlighted.

Although no new compounds are presented in this paper, the optimization of the described synthesis has been carried out very thoroughly therefore the results may have some benefit for other synthetic chemists. The results of biological tests do not show significant activity nor selectivity, but may be the basis for further research. The following minor errors should be corrected:

Line 46 - When first mentioned in the text full Latin names should be given, e.g. Candida albicans, then abbreviated C. albicans.

Table 3. Entry 4, in the 3,Ar column it should be written d not c.

Scheme 2. Arrows should be added to move electrons from the double bond of carbonyl group to oxygen (where is nucleophilic attack at the carbonyl carbon).

Table 4. The unit is missing (mM?).

Author Response

Ad. 1. Full latin names of the mentioned microorganisms have been given.

Ad. 2. In Table 3, entry 4, "c" has been replaced by "d".

Ad. 3. The arrows in the scheme have been added

Ad. 4. The units were added in Table 4.

Round 2

Reviewer 2 Report

The optimization of the reaction conditions does not present a great relevance that requires  to be described in a new article.

The cytotoxic activity of the synthesized compounds is not showing good results.

Perhaps it would be convenient to study other possible biological activities  and/or structural modifications that allow a better biological response.

Author Response

We agree, that the optimization performed was very simple, but it produced good to very good results. For example, the yield of, 2-(4,6-dibromo-1H-benzimidazol-1-yl)-1-(2,4-difluorophenyl)ethanone (5h) increased from 22% to 74%. In case of 2-(4,6-dibromo-1H-benzimidazol-1-yl)-1-(3,4-dichlorophenyl)ethanone (5f) the increase was more than fivefold (15% and 77% after optimization). Isomeric compound, 2-(4,6-dibromo-1H-benzimidazol-1-yl)-1-(2,4-dichlorophenyl)ethanone (5e) was obtained with an almost four times higher yield (21% and 80%). Achieving such yields, we did not see the need for further optimization. Moreover, the methods used are cheap and easy to perform, which in our opinion is an advantage of the presented study. Taking into account the above and the fact that compounds 5e, 5f, and 5h are interesting lead structures in the search for new antifungal API, the research presented is worth publishing. These compounds exhibit high activity against Candida albicans and Candida neoformans and low cytotoxicity against the Vero cell line. In addition, compound 5h demonstrated a significant response against the fungal virulence factors (published in Molecules 2021, 26, 5463. doi.org/10.3390/molecules26185463). The cytotoxic activity of the synthesized compounds is moderate, however, compounds 4f, 5g, and 5j indicate pro-apoptotic properties towards leukemic cells. They demonstrate similar cytotoxicity to 4,5,6,7-tetrabromo-1H-benzimidazole (TBBi) – a known inhibitor of protein kinase CK2. We have added EC50 values for TBBi for all tumor cells (Table 4). Interestingly, as with the compounds tested, A-549 cells are the least sensitive to TBBi (the highest EC50). Of course, we agree, that it is a good idea to investigate further biological activities, e.g. antibacterial and antiviral, but we do not currently have the capacity to do so.

Reviewer 3 Report

Comparing the added positive control in the cytotoxicity testing, there are no comparable results from any synthesized compound, which leads to low significance for the manuscript. I would recommend further exploring more compounds to find at least one with comparable cytotoxicity properties or testing other biological functional properties.

Author Response

As suggested by the reviewer, we have changed the positive control to a compound showing comparable cytotoxic properties to the compounds tested, i.e. 4,5,6,7-tetrabromo-1H-benzimidazole (TBBi). Regarding a similar structure of TBBi to the tested compounds, it should be better positive control in our studies than CX-4945. TBBi is ATP-competitive inhibitor of protein kinase CK2 and has been used by us and others as a reference in anticancer studies for many years. Interestingly, as with the compounds tested, A-549 cells are the least sensitive to TBBi (the highest EC50). We have added EC50 values for TBBi for tumor cells instead of CX-4945 (Table 4). We have also added the representative graph for A-549 cell line in Supporting Info (Figure S3).